# Effect of Al–Li Alloy on the Combustion Performance of AP/RDX/Al/HTPB Propellant

Weiqiang Xiong [1,2], Yunjie Liu [2], Tianfu Zhang [2], Shixi Wu [2], Dawen Zeng [1,*], Xiang Guo [2] and Aimin Pang [3]

1    School of Materials Science and Engineering, Huazhong University of Science and Technology, Wuhan 430074, China
2    Science and Technology on Aerospace Chemical Power Laboratory, 58 Qinghe Road, Xiangyang 441003, China
3    The Fourth Academy of CASC, Xi'an 710025, China
*    Correspondence: dwzeng@mail.hust.edu.cn

**Abstract:** Aluminium–lithium alloy (Al–Li alloy) powder has excellent ignition and combustion performance. The combustion product of Al–Li alloy powder combined with ammonium perchlorate is gaseous at the working temperature of solid rocket motors, which greatly reduces the loss of two-phase flow. Experimental investigations were thoroughly conducted to determine the effect of the Al–2.5Li (2.5 wt% lithium) content on propellant combustion and agglomeration based on thermogravimetry-differential scanning calorimetry, heat combustion, laser ignition, combustion diagnosis, a simulated 75 mm solid rocket motor and a condensed combustion products (CCPs) collection device. The results show that the exothermic heat and weight gain upon the thermal oxidation of Al–Li alloy is obviously higher than those of Al powder. Compared with the reference propellant's formulation, Al–2.5Li leads to an increase in the burning rate and a decrease in the size of the condensed combustion products of the propellants. As the Al–2.5Li alloy content gradually increases from 0 wt% to 19 wt%, the burning rate increases from $5.391 \pm 0.021$ mm/s to $7.244 \pm 0.052$ mm/s at 7 MPa of pressure; meanwhile, the pressure exponent of the burning rate law is changed from $0.326 \pm 0.047$ to $0.483 \pm 0.045$, and the $d_{43}$ of the combustion residue is reduced from $165.31 \pm 36.18$ μm to $12.95 \pm 4.00$ μm. Compared to the reference propellant's formulation, the combustion efficiency of the HTPB propellant is increased by about 4.4% when the Al–2.5Li alloy content is increased from 0 to 19%. Therefore, Al–2.5Li alloy powder is a promising fuel for solid propellants.

**Keywords:** Al–Li alloy; combustion performance; AP/RDX/Al/HTPB propellant





## 1. Introduction

Aluminum (Al) is one of the most common components in space propulsion. It improves the energy performance of the propellant as a metal fuel because of its high energy density, high combustion temperature, non-reactivity during mixing and storage, environmentally benign by-products, and relatively low cost [1–3]. However, micro-scale Al particles exhibit long ignition delays and relatively slow combustion rates [4,5], resulting in incomplete combustion and ignition failure in some cases [6,7]. Additionally, the performance of solid rocket motors is affected by the combustion of Al particles. With the solid propellant burning, the Al powder will melt on the burning surface of the propellant, forming condensed combustion products (CCPs) with larger sizes [8,9]. These CCPs can deposit on nozzle surfaces, resulting in increased two-phase flow losses in rocket motors [10].

To improve the combustion efficiency of aluminum and reduce two-phase flow losses, extensive research has been conducted on the design of solid rocket motors and the modification of aluminum. The combustion efficiency of the propellant can be improved by increasing the working pressure of the solid rocket motor [11–13]. Nano-aluminum powder can reduce the ignition delay time and improve the energy release efficiency

of the propellant [14,15]. However, the nano-crystallization and surface modification of aluminum lead to losses in the overall energy properties of the propellant [16–21] and therefore have not been applied in practice. Aluminum alloying has great advantages in reducing ignition delay and improving the combustion efficiency [22–24]. The combustion heat of lithium (Li) is 43.1 MJ/kg, which is 39.5% higher than that of aluminum. In addition, lithium has a low melting point and excellent ignition performance [25]. Moreover, lithium and aluminum can form alloys that produce a micro-explosion effect, which is conducive to decreasing the size of combustion products, reducing the loss of two-phase flow, and improving the combustion efficiency of solid rocket motors. Therefore, replacing aluminum with an aluminum–lithium alloy can, in theory, further improve the energy level of solid propellants.

Recent work [26] has shown that there are performance benefits to using Al–20Li alloy (20 wt% lithium) as an ammonium perchlorate composite propellant (APCP) fuel additive. Thermo-chemical calculations showed that using Al–20Li alloy can reduce hydrochloric acid formation by more than 95% and increase the theoretical specific impulse (ISP) by about 7 s compared to neat aluminized APCP [27,28]. The oxidization rate of Al–Li alloy powders is higher than pure Al, owing to the decreasing activation energy caused by the catalysis of the soluble Li in Al and Al–Li compounds. The shorter ignition delay and higher reaction rate of Al–Li/KP with increasing Li content in Al–Li alloy powders is a result of the micro-explosion of Al–Li [29]. Compared with pure Al, Al–3Li alloy powder and Fe/Al–3Li composite powder both exhibit significantly improved thermal reaction activities, which include huge increases in the mass gain and intensive heat release [30].

Although Al–Li alloy powder shows great promise for improving both energetic performance and environmental impact, a quantitative assessment of its combustion performance is needed to properly compare it to other propellants. Firstly, the influence of the Al–Li alloy content on the combustion heat of HTPB propellants is unclear. Moreover, the effect of the Al–Li alloy on the combustion and agglomeration of solid propellants is still lacking and not completely understood. The performance of an aluminized lithium alloy propellant in solid rocket motors needs to be further verified. It is the objective of this work to investigate the thermal and combustion performance of Al–2.5Li binary alloys in APCP formulations using high-speed photography, CCP collection methods and simulated 75 mm solid rocket motors. These results provide theoretical guidance for further application of Al–Li alloy powder in solid propellants with high aluminum content.

## 2. Experimental

### 2.1. Materials

The experiments were based on HTPB propellant. The major components used in this study include aluminum, aluminum–lithium alloy, AP, RDX, additives and an HTPB binder. The composition of the formulation is as follows: 19/59.5/10/11.5 wt% metal additive/AP/RDX/HTPB. The aluminum powder used in this study was purchased from Angang Group Aluminum Powder Co., Ltd., China. The average diameter of Al particles in the microspheres is about 19 μm. The Al–Li alloy powder with the composition of 97.5% Al and 2.4% Li in weight was supplied by the Jiangsu ZhiRen Jing Xing New Material Research Institute. The purity and particle size of the Al–2.5Li alloy are 99.9% and 17 μm, respectively. The RDX was sourced from Gansu Baiyin Chemical Industry Co., Ltd, China. The average size of the RDX particles is about 18 μm. AP was provided by North Potassium Chlorate Industry of Dalian and is divided into two types of particles size with a coarse-to-medium ratio of 1.48:1. The AP was ground in a pulverizer and sieved to obtain the required particle size of 410 μm for coarse particles and 150 μm for medium particles. The hydroxyl-terminated poly-butadiene is provided by Liming Chemical Research Institute, China. The average molecular weight and the hydroxyl content of HTPB are 3000 g/mol and 0.5 mmol/g, respectively.

## 2.2. Propellant Sample Preparation

The propellant ingredients (Table 1) are then weighed using a digital balance (the least count 0.1 g). The first step of the propellant preparation process is to mix the HTPB and Al (Al–2.5Li) well. AP and RDX are added three times and mixed in the vertical kneader at 50 °C. The curing agent is added at last and mixed for about 40 min. The propellant slurry thus obtained after the mixing process is immediately cast. The propellant slurry is then cast into rectangular metal moulds. The cast propellants are placed in a constant temperature hot air oven at a temperature of $50 \pm 2$ °C for 7 days.

**Table 1.** Formulation of prepared propellants.

| Sample No. | HTPB,% | AP,% | RDX,% | Al,% | Al–2.5Li,% |
|---|---|---|---|---|---|
| HA-1 | 11.5 | 59.5 | 10.0 | 19.0 | 0.0 |
| HA-2 | 11.5 | 59.5 | 10.0 | 14.0 | 5.0 |
| HA-3 | 11.5 | 59.5 | 10.0 | 9.0 | 10.0 |
| HA-4 | 11.5 | 59.5 | 10.0 | 5.0 | 15.0 |
| HA-5 | 11.5 | 59.5 | 10.0 | 0 | 19.0 |

## 2.3. Equipment and Experimentation

### 2.3.1. Thermal Test

The thermal properties of the Al/Al–2.5Li alloy composite were studied using simultaneous thermo-gravimetric analysis (TG) and differential thermal analysis (DTA) using an instrument (NETZSCH, STA449F3) with 50 mL/min of oxygen flow and in temperatures from 40 °C to 1400 °C at a heating rate of 20 °C/min.

### 2.3.2. Combustion Heat Test

A certain amount of solid propellant was preset in the oxygen bomb calorimeter (Parr 6200, Parr Instrument Co., Moline, IL, USA) with an argon pressure of ~3.0 MPa. The propellant was ignited by an electrically heated nickel–chromium wire [31] and burned in the oxygen bomb calorimeter. Based on the measured increase in temperature of the water in the inner cylinder, the total heat of the process was calculated using the water equivalent of the calibrated calorimeter system. Then, the heat of the combustion of the propellant sample was calculated from the total heat.

### 2.3.3. Burning Rate Test

The burning rate was measured using an industry-standard acoustic emission technique. The propellant samples are cut into 5.0 mm × 5.0 mm × 84.8 mm strands, coated with polyvinyl alcohol, and aired five times. A fine metal wire was threaded through the top of the samples to ignite the propellant using a voltage at the initial temperature of 20°C in the stellar bomb, which was filled with a nitrogen atmosphere and immersed under water. In order to obtain the burning rate, two other low-melting-point fine fuse wires were threaded through the strand at separate distances of 100 mm to record the start and end time signals of the combustion. The real-time data were recorded by a computer to process and calculate the burning rate. Five replications of the combustion were performed under each pressure (3.0, 5.0, 7.0, 9.0 MPa), and the data were averaged [32].

### 2.3.4. Combustion Diagnostics System

In order to study the microscopic behavior of aluminum agglomerates on the burning surface of the propellant under spontaneous combustion, the same combustion diagnostic apparatus as in [33] was used. Combustion experiments were conducted in a nitrogen atmosphere at room temperature. The initial high-pressure environment was established by filling the combustion chamber with nitrogen provided by the tank. The combustion and agglomeration characteristics were observed by a high-speed camera (Phantom M340, USA) at 8400 frames per second through a long-range microscope. The exposure time per

frame was set to 30–45 µs, depending on the chamber pressure and the flame intensity of the propellant samples. Combustion tests were conducted at 5 MPa pressure.

### 2.3.5. Condensed Combustion Products (CCPs) Test

The experimental CCPs collection system mainly included the product collection device, pressure data acquisition system, pressurization system and exhaust system. The collection device was made of carbon steel. The length of the tube in the combustion chamber was 195 mm. The collection chamber is removable, and its length is constant. The principle of this device is that the amount of gas produced by the propellant is equal to the amount of gas emitted by the exhaust device. Further details can be found in [34].

The particle size distribution of CCPs was measured by a laser diffraction particle size analyzer (Malvern Mastersizer 2000). The CCPs used for particle size testing were in dry powder form and were sonicated prior to testing. The parameter of ultrasonic dispersion was supposed to be 40 W, 5 min before size measurement. The carrier liquid is pure water.

### 2.3.6. The simulated 75 mm Test Solid Rocket Motor

The combustion efficiency of aluminum powder was tested in a simulated 75 mm test solid rocket motor with a 4.5 mm throat neck. The simulated 75 mm test solid rocket motor is closed at the bottom and open at the upper end, with an inside diameter of 65 mm, an outside diameter of 75 mm. The mass of solid propellant charge in Φ75 mm is 220 g, and the test run temperature is 20 °C. According to the change in mass before and after the simulated 75 mm solid rocket motor test, the combustion efficiency of aluminum powder added to the propellant formulation can be calculated by the following formula:

$$\eta(Al) = \frac{M_1 - m_1 - m_2}{M} \times 100\% \qquad (1)$$

where $M_1$ is the mass of the simulated 75 mm solid rocket motor before the test, g. $m_1$ is the mass of the simulated solid rocket motor after testing, and g. $m_2$ is the mass of the ignition charge.

In order to investigate the reaction degree of aluminum powder in the propellant, the active aluminum content in the test residue of the simulated 75 mm test solid rocket motor was measured. Aluminum reacts with sodium ethylenediaminetetraacetic acid disodium solution at pH 2.5–2.8. When the pH value is 5–6, excess EDTA is titrated with zinc chloride standard titration solution, and then fluoride is used to react with aluminum and release a certain amount of EDTA. This reaction consumes the volume of zinc chloride in the standard titration solution so the aluminum content can be calculated [35]. Therefore, the aluminum content in propellant combustion residues can be calculated by the following formula:

$$W(Al) = \frac{(V/1000)CM_2}{\frac{m \times 25}{250}} \times 100 \qquad (2)$$

where $V$ is the volume of the consumed zinc chloride standard titration solution, in ml; $C$ is the concentration of zinc chloride standard titration solution, in mol/L; $M_2$ is the molar mass of aluminum, in g/mol; and m is the mass of the sample, in g.

## 3. Results and Discussion

### 3.1. Combustion Heat

It is well-known that large aluminum agglomerates can reduce the velocity of the surrounding gas product, resulting in the incomplete thermal energy conversion of solid propellants containing aluminum, thereby reducing their actual thermal properties. The evolution of the combustion heat in a calorimetric bomb by the deflagration of a propellant in an inert gas is also often used to preliminarily evaluate the thermal energy conversion of the propellant [36]. As listed in Table 2, the combustion heat of the propellant formulation increases gradually with the increase in the Al–2.5Li alloy content and reaches the maximum value when Al–2.5Li alloy content is 19%; moreover, the combustion heat of

the Al–2.5Li/AP/RDX/HTPB-propellant is increased by about 200 J/g. The combustion heats of the Al–2.5Li alloy, Al and their mixtures are shown in Table 3. The combustion calorific value of Al–2.5Li alloy is about 700 J/g higher than that of Al powder. The theoretical calorific value of lithium is significantly higher than that of aluminum powder. The calorific value of lithium and aluminum alloy is higher than that of pure aluminum, and the reactivity of Al–2.5Li alloy is higher than that of aluminum powder. The higher reaction activity enables the rapid completion of the combustion reaction. Therefore, the combustion heat of Al–2.5Li alloy is higher than that of aluminum powder.

**Table 2.** Combustion heat of HTPB propellant containing Al–2.5Li alloy.

| Sample No. | Al, % | Al–2.5Li, % | Combustion Heat (J/g) |
|---|---|---|---|
| HA-1 | 19 | 0 | 6167.0 ± 44.5 |
| HA-2 | 14 | 5 | 6196.9 ± 49.0 |
| HA-3 | 9 | 10 | 6238.6 ± 61.0 |
| HA-4 | 4 | 15 | 6273.4 ± 55.0 |
| HA-5 | 0 | 19 | 6310.5 ± 59.5 |

**Table 3.** Combustion heat of Al–2.5Li alloy, Al and their mixtures.

| Sample No. | Al, % | Al–2.5Li, % | Combustion Heat (J/g) |
|---|---|---|---|
| X-1 | 19 | 0 | 28880.20 ± 40.10 |
| X-2 | 14 | 5 | 29172.26 ± 39.27 |
| X-3 | 9 | 10 | 29348.80 ± 35.86 |
| X-4 | 4 | 15 | 29442.84 ± 52.17 |
| X-5 | 0 | 19 | 29598.41 ± 39.65 |

*3.2. Thermal Characteristics*

In order to further investigate the oxidization mechanism of Al–2.5Li alloy, the thermal behaviors of the Al–2.5Li alloy, Al powder and their mixtures in oxygen were studied by TG/DTA. DTA tracing of the sample in $O_2$ is shown in Figure 1b. First of all, it can be seen from DTA curve, a weak endothermic peak appears in all samples at 660 °C, and there is no obvious change in the TG curve at this temperature. It is not difficult to infer that the melting transformation of the aluminum matrix occurs at this time. When the temperature reaches 1100°C, the Al–2.5Li alloy fuel shows an obvious heat release phenomenon during oxidation, the DTA curve shows an obvious concentrated exothermic peak, and the TG curve shows an obvious weight gain phenomenon. The exothermic peak temperature of pure Al–2.5Li alloy powder and pure Al powder are 1076.8°C and 1069.8°C, respectively, but the exothermic ability of Al–2.5Li alloy powder is higher than that of Al powder, which indicates that the Al–2.5Li alloy has a higher activity and thermal oxidation reaction than Al powder. Figure 1a shows the TG curves of a physical mixture of an Al–2.5Li alloy and aluminum powder having different contents. Based on the TG curve shown in Figure 1a, the observed weight change is consistent with the exothermic effect shown in the DTA curve in Figure 1b. The weight gain seen in Al–2.5Li alloy upon thermal oxidation is about 1.6 times that of Al. Compared with the slow and incomplete oxidation of elemental aluminum, the oxidation of Li-doped Al–Li alloy fuel is stronger and more thorough. In fact, for the basic Al feedstock, the dense oxide film on the particle surface will hinder the contact of oxygen with the internal active Al, thus limiting the further oxidation of Al. In contrast, the preferential oxidation of Li may provide more channels for the contact between oxygen and Al and promote the full combustion of Al–2.5Li alloy fuel.

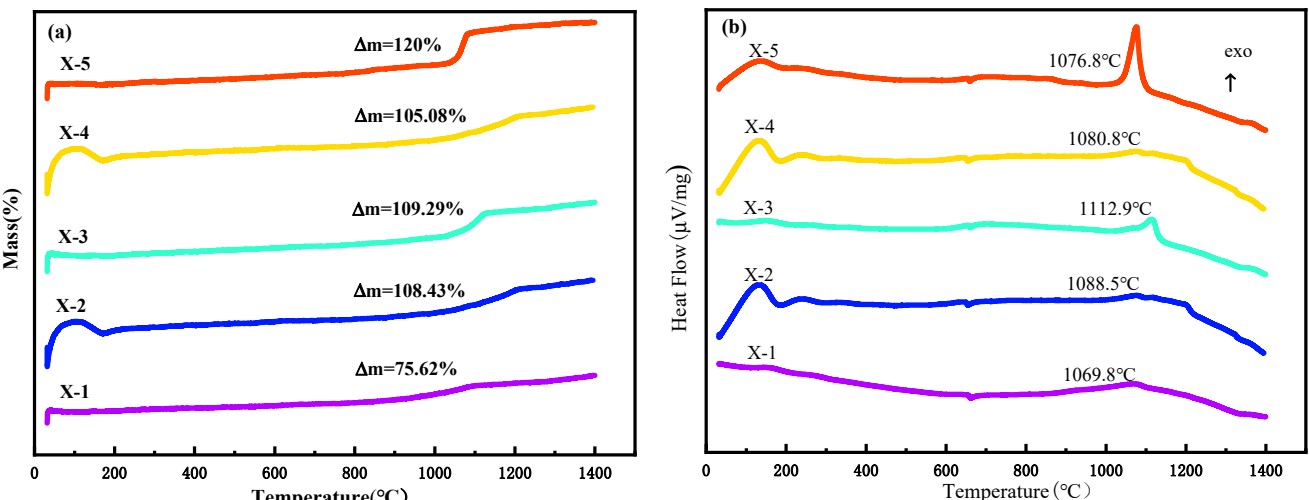

**Figure 1.** DTA curves and TG curves of Al–2.5Li alloy: (**a**) TG curves; (**b**) DTA curves. Al powder and their mixtures at a heating rate of 20 °C/min in oxygen.

### 3.3. Combustion Characteristics

As expected, Al–2.5Li alloy affects the combustion characteristics of solid propellants. The burning rate of a solid propellant is closely related to the pressure [37]. Combustion rate data (Figure 2) at different pressures are also presented according to a similar method given in Ref [38], and the pressure of the combustion rate is calculated by quantizing the combustion rate coefficient a and pressure exponent n according to Vieille's law ($r = ap^n$).

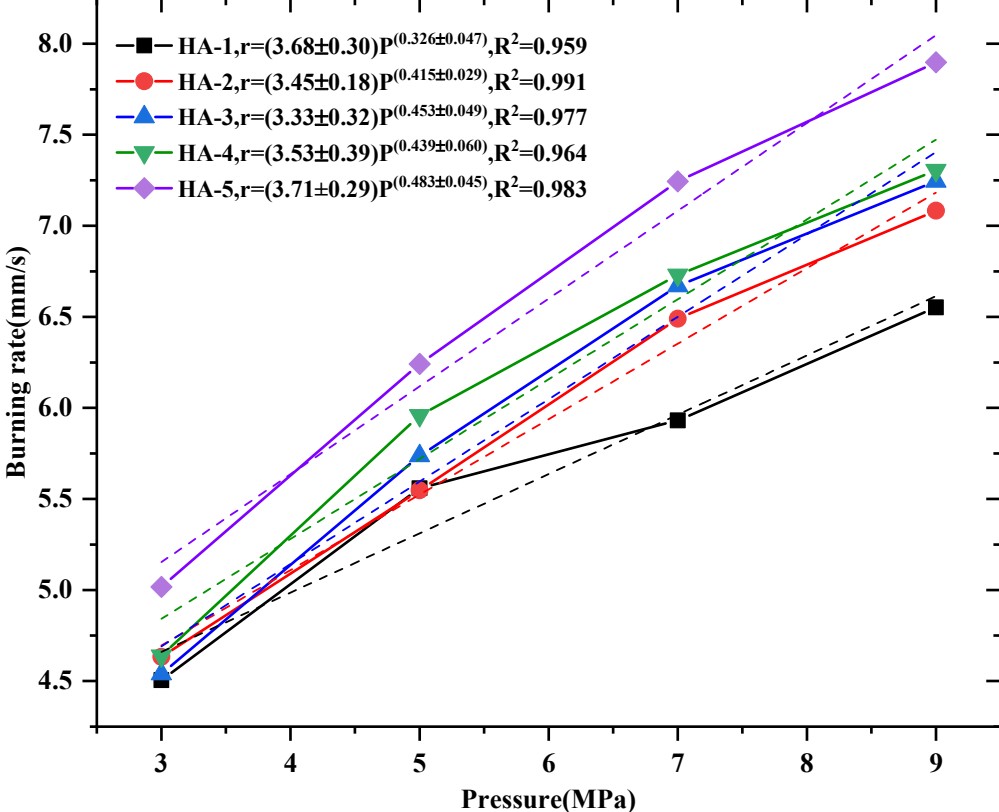

**Figure 2.** Linear burning rate and power-law pressure dependence in 3 MPa–9 MPa for solid propellants.

As the Al–2.5Li alloy content gradually increases from 0 wt% to 19 wt%, the burning rate increases from 5.391 mm/s to 7.244 mm/s at a pressure of 7 MPa. The pre-exponent of burning rate law is changed from 3.68 to 3.45 while the pressure exponent increases from 0.326 to 0.415 with partial replacement of 5 wt% Al by Al–2.5Li. As the content of the Al–2.5Li alloy in the propellant enhances to 19 wt%, the pre-exponent reaches 3.71, whereas the pressure dependency increases to 0.483. The results indicate that the content of Al–2.5Li alloy has a great effect on the burning rate of propellants. The addition of the Al–2.5Li alloy leads to the increase in the burning rate and pressure exponent of the propellant's formulation. This higher-pressure sensitivity may be explained by the increased micro-explosion effect of the smaller Al–2.5Li particles being a dominant driver in the burning rate at low pressure. Li in the Al–2.5Li alloy could evaporate rapidly during the combustion, resulting in micro-explosions, which disperses the alloy powders into small particles. The small particles can increase the interface area and effective mass diffusion rate. A micro-explosion produces an atomized fuel mist that burns in a more pressure-sensitive, kinetically controlled manner rather than a diffusion-controlled manner [39].The work of Ao's team [38] also revealed a similar mechanism of action of aluminum alloy in propellant combustion.

To gain a deep insight into the combustion process of Al–2.5Li alloy in solid propellant, laser ignition of propellant powder has been studied. Figure 3a shows the typical agglomeration process of aluminum particles, including ignition combustion and agglomeration. The final agglomeration is characterized by spherical droplets of liquid metal having a diameter of about 950 μm. Compared with aluminum powder, the combustion of Al–2.5Li alloy powder is relatively complete, and there is no serious agglomeration under the combustion chamber pressure of 5 MPa. The results of previous experiments show that the ignition reaction temperature of aluminum powder is relatively high, which is close to 660 °C. For composite solid propellants, ignition mainly depends on the high temperature of ammonium perchlorate (AP) and the diffusion flame of the binder. Therefore, the ignition combustion zone of the aluminum powder is remote from the combustion surface, and the ordinary aluminum powder tends to condense on the combustion surface of the propellant to form large aluminum agglomerates which tend to ignite and burn away from the combustion surface (Figure 3a). At this time, the energy fed back to the combustion surface of the aluminum powder is correspondingly reduced. Lithium and aluminum alloys have lower melting points than pure aluminum. Compared with pure aluminum, the reaction activity is greatly improved. At the low temperature of 500~600 °C, the reaction is rapid, and the oxidation heat release, ignition, and combustion occur near the combustion's surface. Thermal feedback is increased, and the combustion of the propellant is promoted. Therefore, the higher combustion rate and reaction activity rapidly cause the Al–2.5Li alloy to leave the combustion surface of the propellant and diffuse into the gas phase, thereby reducing the formation of CCPs.

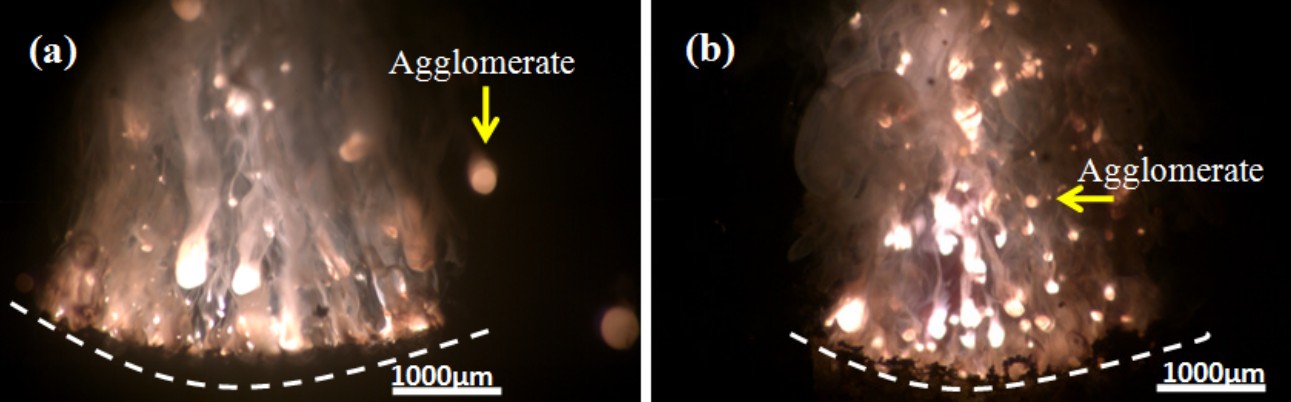

**Figure 3.** Combustion of propellant at 5 MPa pressure: (**a**) HA-1; (**b**) HA-5.

### 3.4. Condensed Combustion Products

To further understand the high combustion efficiency of Al–2.5Li alloy, the particle size distribution of the propellant's combustion residue was studied. The effect of the particle size distribution of CCPs of different Al–2.5Li content in HTPB propellants at pressure 5 MPa is shown in Figure 4. As shown in Figure 4, the size range of the CCPs is 0.02–2000 μm. The particle size distribution of condensed-phase combustion products of aluminum powder formula is mainly in the range of 100 ~ 1000 μm. With the increase in the amount of Al–2.5Li alloy powder added in the solid propellant formulation, the particle size distribution of the condensed phase combustion product shifts to the left, and the particle size decreases gradually. The average particle sizes $d_{43}$ of the condensed combustion products of the five propellant formulas are 165.31 μm, 90.06 μm, 57.48 μm, 22.32 μm and 22.32 μm, respectively. Obviously, the particle size of the Al–2.5Li-containing propellant is significantly lower than that of the reference propellant. The average particle size $d_{43}$ of the CCPs of the HTPB propellant decreases sharply when the Al–2.5Li content is in the range of 0~19 wt%. When the content of Al–Li alloy is 19%, the relative content of agglomerates larger than 160 μm to the CCPs of the combustion products is very small. This indicates that, with the increase in the Al–Li alloy content, the content of large-sized aggregates decreases. Based on the image of the combustion's surface, when the content of Al–Li alloy is 19%, the average particle size of the CCP is greatly reduced. A possible reason for this is the high content of aluminum powder in the reference propellant's formulation, which by itself has an incomplete combustion phenomenon. Al–2.5Li alloy produces a micro-explosion effect during combustion, and the combustion product of lithium can react with hydrogen chloride to generate lithium chloride, which is beneficial in reducing the formation of condensed phase products, thus reducing the particle size of the residue from thermal combustion [40]. In addition, the aluminum–lithium alloy powder has a higher heat release and a lower ignition combustion temperature, which reduces the formation of CCPs. Higher burning rates can also inhibit the formation of agglomerates.

### 3.5. Demonstration of the Simulated 75 mm Test Solid Rocket Motor

In order to further understand the effect of Al–2.5Li alloy on the combustion efficiency of the HTPB propellant, the combustion performance of the HTPB propellant containing Al–2.5Li alloy was verified using a simulated 75 mm test solid rocket motor. The results are shown in Table 4 and Figure 5. Figure 5b shows the nozzle throat slag of the simulated solid rocket motor using a reference formulation. The large deposits of slag at the end of the combustion chamber indicate that during propellant combustion, a large number of agglomerated particles collide with the inner wall of the combustion chamber in this region. Compared to Figure 5a, the agglomeration observed when using the aluminum powder formulation was more severe and the size of the agglomerated residue was larger. As shown in Table 4, the blank formulation had the lowest combustion efficiency and the highest active aluminum content in the combustion residue. However, Al–2.5Li alloy formulations show opposite results. Compared with the blank formulation, when the content of Al–2.5Li alloy is 19%, the combustion efficiency of the propellant formulation is increased by 4.4% and the active aluminum content in the combustion residue decreases by 5.7%. With the increase in Al–2.5Li alloy content, the combustion efficiency of the propellant increases gradually, and the active aluminum content in the combustion residue decreases gradually. This proves again that Al–2.5Li alloy can improve combustion efficiency of HTPB propellant with high Al content.

**Table 4.** Combustion efficiency of Al/Al–2.5Li powder and fraction of residual active aluminum of propellants.

| Sample No. | Combustion Efficiency ($\eta\%$) | Fraction of Residual Active Aluminum (wt.%) |
|---|---|---|
| HA-1 | $92.273 \pm 0.455$ | $8.39 \pm 0.10$ |
| HA-2 | $93.636 \pm 0.376$ | $6.95 \pm 0.08$ |
| HA-3 | $94.091 \pm 0.154$ | $5.87 \pm 0.09$ |
| HA-4 | $95.545 \pm 0.167$ | $4.46 \pm 0.07$ |
| HA-5 | $96.600 \pm 0.223$ | $2.65 \pm 0.05$ |

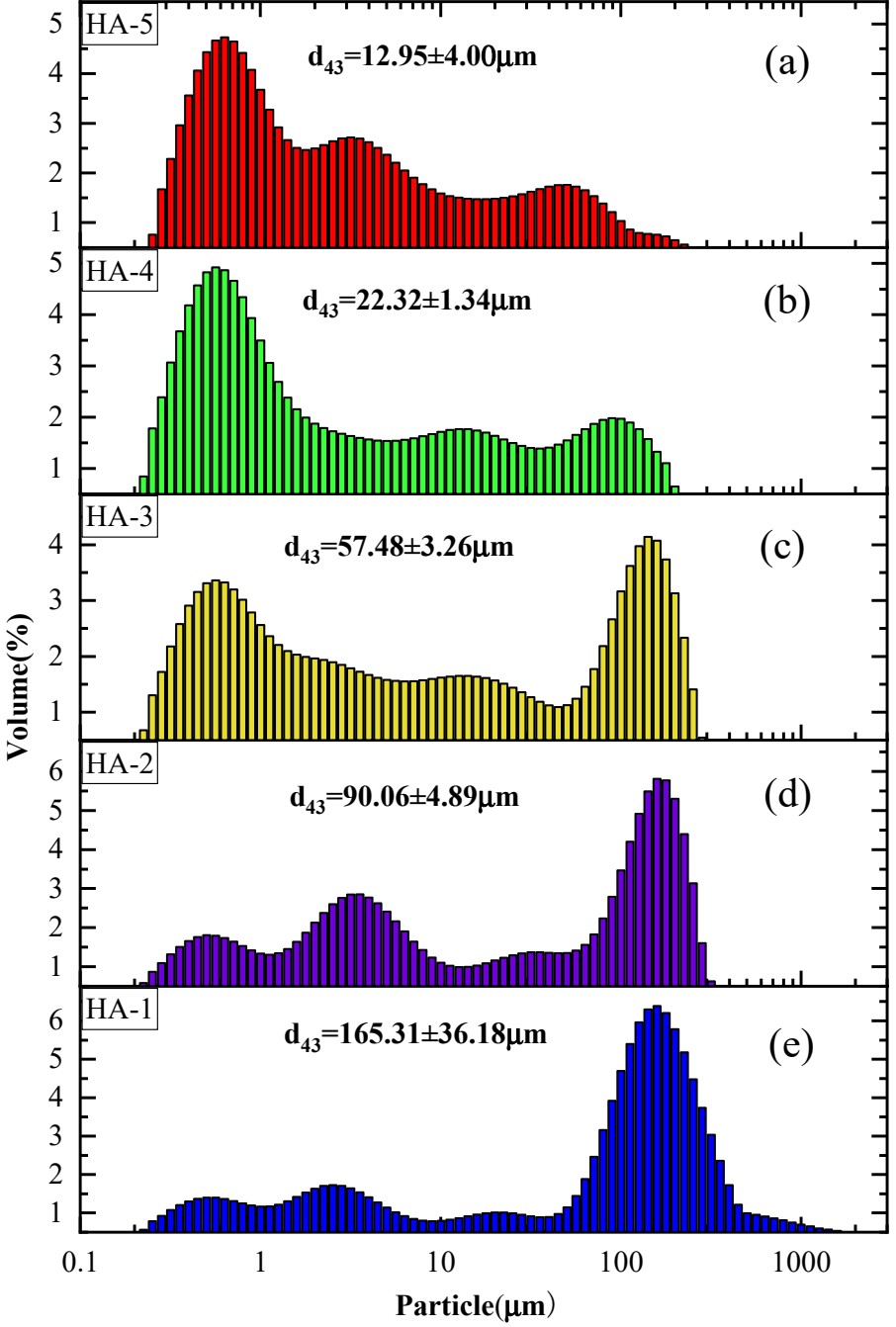

**Figure 4.** Size distribution of the combustion residue of the propellant: (**a**) HA-5; (**b**) HA-4; (**c**) HA-3; (**d**) HA-2; (**e**) HA-1.

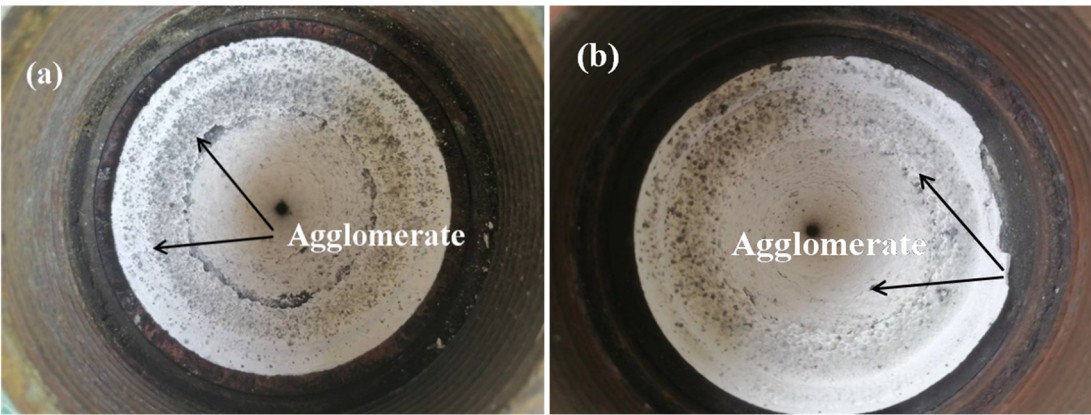

**Figure 5.** Propellant combustion residue deposit at nozzle of 75 mm simulated test solid rocket motor: (**a**) Al–2.5Li/AP/RDX/HTPB propellant (HA-5); (**b**) Al/AP/RDX/HTPB propellant (HA-1).

## 4. Conclusions

In this study, the effect of Al–2.5Li content on the combustion of HTPB solid propellants was investigated. The combustion characteristics of the HTPB propellant with high aluminum content were analyzed via laser ignition, the condensed combustion products (CCPs) collection method and a simulated 75 mm test solid rocket motor. The influence mechanism of the Al–2.5Li alloy content on the agglomeration and combustion of HTPB propellant was revealed. Al–2.5Li alloy can reduce particle agglomeration in the combustion of HTPB propellants and improve combustion intensity and combustion efficiency of propellant formulations. For pressures in the range of 3–9 MPa, the Al–2.5Li alloy content has a great influence on the burning rate and the burning rate pressure exponent. With the increase in the Al–2.5Li alloy content, the pressure exponent and burning rate increase. The degree of combustion residue agglomeration and the active aluminum content in the residue of the HTPB propellant decrease with the increase in the Al–2.5Li alloy content. In conclusion, the content of Al–2.5Li alloy has great influence on the agglomeration and combustion characteristics of HTPB propellants. Al–Li alloys have potential applications in solid propellants.

**Author Contributions:** W.X.: conceptualization, investigation, methodology, formal analysis, writing, data curation, software, and preparation. Y.L.: original data curation, draft preparation, visualization, and validation. T.Z.: methodology and data curation. S.W.: resources and validation. D.Z.: methodology and formal analysis. X.G.: review and editing, visualization, funding acquisition, and project administration. A.P.: project administration and supervision. All authors have read and agreed to the published version of the manuscript.

**Funding:** This research was funded by the National Natural Science Foundation of China (NSFC21905084, U20B2018) and the Open Foundation of the Science and Technology on Aerospace Chemical Power Laboratory (STACPL320201B02, STACPL320181B03-1 and STACPL220221B02).

**Institutional Review Board Statement:** Not applicable.

**Informed Consent Statement:** Not applicable.

**Data Availability Statement:** Not applicable.

**Acknowledgments:** The authors are grateful to the National Natural Science Foundation of China and the Open Foundation of the Science and Technology on Aerospace Chemical Power Laboratory for financial support of this work.

**Conflicts of Interest:** The authors declare no conflict of interest.

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
