# Peer review of "Effect of Al–Li Alloy on the Combustion Performance of AP/RDX/Al/HTPB Propellant"

_aerospace, doi:10.3390/aerospace10030222_

Round 1

Reviewer 1 Report

I read the manuscript entitled "Effect of Al-Li Alloy on Combustion Performance of AP/RDX/Al/HTPB Propellant" with interest. It is a thorough study reporting the interesting results for aluminum-lithium alloys (emerging topic in propellants). I suggest acceptance after the slight polishing of text by authors. Comments from the beginning:

abstract: "increases from 5.391mm/s to 7.244mm/s at pressure 7MPa, meanwhile the pressure exponent of burning rate law is changed from 0.326 to 0.483, and the d43 of the combustion residue is reduced from 165.310μm to 12.951μm" - please, provide the errors for all these values

introduction - the topic of Al combustion is well studied, and to give a broader overview and more focused problem statement i suggest to discuss the following literature in addition: https://doi.org/10.1021/nl025905k,   https://doi.org/10.1002/prep.201000028 (nanosize/mixing effects)

introduction: "Recent work[22] has shown that there are performance benefits to using Al-20Li alloy (20wt% lithium) as an ammonium perchlorate composite propellant(APCP) fuel additive" - should be Ref.23, not 22

"HTPB Propellants" -> "propellants with HTPB binder"

section 2.3.1 "20°C min." -> "20°C min^-1."

section 2.3.3 "3,5,7,9MPa" -> "3.0, 5.0, 7.0, 9.0 MPa"

"2.3.4. combustion" -> "2.3.4. Combustion"

section 2.3.6 "in Φ75mmis" - what is it?

section 3.1 "Although the caloric value of aluminum and lithium alloy will decrease after alloying with lithium" - it is not clear

Figure 1 - TG curves are shown in (a) plot, please correct it in text and figure caption

Figure 2 - it is more common to represent the combustion results in log-log plot

Author Response

Response to Reviewer 1 Comments

Title: Effect of Al-Li Alloy on Combustion Performance of AP/RDX/Al/HTPB Propellant

Manuscript ID: aerospace-2179487

Dear Reviewer,

Thank you very much for the kind and helpful comments, which are very important to improve the quality of our work. Taking all the comments into consideration, we carefully and thoroughly modified the manuscript and supporting information. Our point-by-point responses to the comments are as follows and the changes are all marked in red in the revised documents.

REVIEWER REPORT(S):

Reviewer 1

Comments to the Author:

  1. abstract: "increases from 5.391mm/s to 7.244mm/s at pressure 7MPa, meanwhile the pressure exponent of burning rate law is changed from 0.326 to 0.483, and the d43 of the combustion residue is reduced from 165.310μm to 12.951μm" - please, provide the errors for all these values.

Response: Thank you very much for your comment. The abstract was revised correspondingly. The details are as following.

The burning rate increases from 5.391±0.021mm/s to 7.244±0.052mm/s at 7MPa of pressure; meanwhile, the pressure exponent of the burning rate law is changed from 0.326±0.047 to 0.483±0.045, and the d43 of the combustion residue is reduced from 165.31±36.18μm to 12.95±4.00μm.

  1. introduction - the topic of Al combustion is well studied, and to give a broader overview and more focused problem statement i suggest to discuss the following literature in addition: https://doi.org/10.1021/nl025905k,https://doi.org/10.1002/prep.201000028 (nanosize/mixing effects)

Response: Thank you very much for your comment. The introduction was also revised correspondingly, and the corresponding references were cited. The details are as follows.

Nano-aluminum powder can reduce the ignition delay time and improve the energy release efficiency of the propellant [14,15]

References

  1. Armstrong, R.; Baschung, B.; Booth, D.; Samirant, M. Enhanced propellant combustion with nanoparticles. Nano Lett. 2003, 3, 253-255.
  2. Yilmaz, N.; Donaldson, B.; Gill, W. Aluminum Particle Ignition Studies with Focus on Effect of Oxide Barrier. Aerosp. 2023, 10, 45.

  1. Introduction: "Recent work [22] has shown that there are performance benefits to using Al-20Li alloy (20wt% lithium) as an ammonium perchlorate composite propellant(APCP) fuel additive" - should be Ref.23, not 22

Response: Thank you very much for your comment. The introduction had been amended accordingly to change the order of citation, the details as detailed below.

Recent work [26] has shown that there are performance benefits to using Al-20Li alloy (20wt% lithium) as an ammonium perchlorate composite propellant (APCP) fuel additive.

References

  1. Terry, B.C.; Sippel, T.R.; Pfeil, M.A.; Gunduz, I.E.; Son, S.F. Removing hydrochloric acid exhaust products from high performance solid rocket propellant using aluminum-lithium alloy. J. Hazard. Mater. 2016, 317, 259-266.

  1. "HTPB Propellants" -> "propellants with HTPB binder"

section 2.3.1 "20°C min." -> "20°C min^-1."

section 2.3.3 "3,5,7,9MPa" -> "3.0, 5.0, 7.0, 9.0 MPa"

"2.3.4. combustion" -> "2.3.4. Combustion"

section 2.3.6 "in Φ75mmis" - what is it?

section 3.1 "Although the caloric value of aluminum and lithium alloy will decrease after alloying with lithium" - it is not clear

Figure 1 - TG curves are shown in (a) plot, please correct it in text and figure caption

Figure 2 - it is more common to represent the combustion results in log-log plot

Response: Thank you very much for your comment. The section two had been amended. The details as detailed below.

section 2.3.1 at a heating rate of 20°C/min.

section 2.3.3 under each pressure (3.0,5.0,7.0,9.0 MPa)

section 2.3.4 Combustion diagnostics system

section 2.3.6 "in Φ75mmis" in Φ75mm is

section 3.1 "Although the caloric value of aluminum and lithium alloy will decrease after alloying with lithium" →The calorific value of lithium and aluminum alloy is higher than that of pure aluminum.

Figure 1 - TG curves are shown in (a) plot, please correct it in text and figure caption→Figure 1.DTA curves and TG curves of Al-2.5Li alloy: (a) TG curves; (b) DTA curves. Al powder and their mixtures at a heating rate of 20℃/min in oxygen.

Figure 2 is redrawn, the details as detailed below.

Reviewer 2 Report

Thank you for this submission. This is a concise, well-written report that reveals interesting effects of the addition of lithium on aluminized propellant combustion. I have only a few minor comments:

- A few acronyms (e.g. TG/DTA) are not explained at their first usage.

- Sections 2.3.2, 2.3.3, and 2.3.5 do not provide enough detail for readers unfamiliar with these experimental techniques to understand how the experiments are performed. Please expand your overview of the techniques described in these sections.

- Results reported in Tables 2-4 seem to have too many significant figures. For example, one combustion heat is 29172.26±39.27 J/kg. Why is the base value reported to within 0.01 J/kg when the uncertainty in the value is much larger than that?

- In Figure 3, the notation "1000 μm" is provided, but I'm not sure what it refers to. Is there meant to be a scale bar?

- Figure 4 has some y-axis values that are hard to read because values from multiple sub-graphs overlap. 

Author Response

Response to Reviewer 2 Comments

Title: Effect of Al-Li Alloy on Combustion Performance of AP/RDX/Al/HTPB Propellant

Manuscript ID: aerospace-2179487

Dear Reviewer,

Thank you very much for the kind and helpful comments, which are very important to improve the quality of our work. Taking all the comments into consideration, we carefully and thoroughly modified the manuscript and supporting information. Our point-by-point responses to the comments are as follows and the changes are all marked in red in the revised documents.

REVIEWER REPORT(S):

Reviewer 2

Comments to the Author:

  1. A few acronyms (e.g. TG/DTA) are not explained at their first usage.

Response: Thank you very much for your comment. The details are as following.

A few acronyms (e.g. TG/DTA) are explained at their first usage.

→section 2.3.1 thermo-gravimetric analysis (TG) and differential thermal analysis (DTA).

  1. Sections 2.3.2, 2.3.3, and 2.3.5 do not provide enough detail for readers unfamiliar with these experimental techniques to understand how the experiments are performed. Please expand your overview of the techniques described in these sections.

Response: Thank you very much for your comment. The Sections 2.3.2, 2.3.3, and 2.3.5 was also revised correspondingly, and the enough details were provided. The details are given in the manuscript.

  1. Results reported in Tables 2-4 seem to have too many significant figures. For example, one combustion heat is 29172.26±39.27 J/kg. Why is the base value reported to within 0.01 J/kg when the uncertainty in the value is much larger than that?

Response: Thank you very much for your comment. The error of the test result of combustion heat is 2%~3%.

  1. In Figure 3, the notation "1000 μm" is provided, but I'm not sure what it refers to. Is there meant to be a scale bar?

Response: Thank you very much for your comment. The notation "1000 μm" meant to be a scale bar.

    5.Figure 4 has some y-axis values that are hard to read because values from multiple sub-graphs overlap.

Response: Thank you very much for your comment. Figure 2 is redrawn, the details as detailed below.

Reviewer 3 Report

This paper needs to be improved with the following recommended changes before being considered for publication. Please see below my comments.

- There are numerous grammatical issues in the paper which should be reviewed by a native speaker accordingly.

- All abbreviations must be given full where first in use.

- Purpose of the study should be stated in a better way to reflect the importance of the work in terms of its application.

- Introduction must be improved to include additional relevant papers in terms of solid rocket propellant burn rate measurements (Propellants, Explosives, Pyrotechnics 33(2), pp.109-117, 2008), uncertainty, safety issues which may occur in low pressure environment or other important parameters in terms of propellant orientation and its relation with combustion characteristics.

- Literature should include recent papers from Aerospace (20229(6), 325; 20229(11), 677; 202310(1), 45)

- There are several commercial codes which are mainly used for propellant combustion modeling. It is recommended that the authors elaborate on that.

- Uncertainty analysis should be performed with the inclusion of experimental data.

- Graphs should have uncertainty bars.

- Conclusions somewhat repeat what was mentioned in the abstract. I recommend the authors include “future work” as well as the importance of this work in relation to practical applications.

Author Response

Title: Effect of Al-Li Alloy on Combustion Performance of AP/RDX/Al/HTPB Propellant

Manuscript ID: aerospace-2179487

Dear Reviewer,

Thank you very much for the kind and helpful comments, which are very important to improve the quality of our work. Taking all the comments into consideration, we carefully and thoroughly modified the manuscript and supporting information. Our point-by-point responses to the comments are as follows and the changes are all marked in red in the revised documents.

REVIEWER REPORT(S):

Reviewer 3

Comments to the Author:

  1. There are numerous grammatical issues in the paper which should be reviewed by a native speaker accordingly.

Response: Thank you very much for your comment. Numerous grammatical issues in the paper in the paper had been corrected by native speakers. For example,

When solid propellant is burning, Al powder will melt on the burning surface of propellant → When a solid propellant is burned, the Al powder will melt on the burning surface of the propellant,

What is more, the effect of Al-Li alloy on the solid propellant combustion and agglomeration still lacks complete understanding. → Moreover, the effect of the Al-Li alloy on the combustion and agglomeration of solid propellants is still lacking and not completely understood.

The burning rates of the propellants were measured at 3 to 9MPa via an industry-standard acoustic emission technique. → The burning rate was measured using an industry-standard acoustic emission technique.

The collection experimental system of the CCPs mainly included product collection device →The experimental CCP collection system mainly included the product collection device

With the addition of Al-2.5 Li alloy →With the increase in the amount of Al-2.5Li alloy powder added in the solid

  1. All abbreviations must be given full where first in use.

Response: Thank you very much for your comment. The abbreviations were given full. (thermo-gravimetric analysis (TG) and differential thermal analysis (DTA).

  1. Purpose of the study should be stated in a better way to reflect the importance of the work in terms of its application.

Response: Thank you very much for your comment. The purpose of the study is stated, reflecting the importance of the work in terms of its application.

Firstly, the influence of the Al-Li alloy content on the combustion heat of HTPB propellants is unclear. Moreover, the effect of the Al-Li alloy on the solid propellant combustion and agglomeration of solid propellants is still lacking and not completely understood. The performance of an aluminized lithium alloy propellant in solid rocket motors needs to be further verified. It is the objective of this work to investigate the thermal and combustion performance of Al-2.5Li binary alloys in APCP formulations using high-speed photography, CCP collection methods and simulated 75 mm solid rocket motors.

  1. Introduction must be improved to include additional relevant papers in terms of solid rocket propellant burn rate measurements (Propellants, Explosives, Pyrotechnics 33(2), pp.109-117, 2008), uncertainty, safety issues which may occur in low pressure environment or other important parameters in terms of propellant orientation and its relation with combustion characteristics.

Response: Thank you very much for your comment. The reference was added to the section 3.3. The details are as following.

As expected, Al-2.5Li alloy affects the combustion characteristics of solid propellants. The burning rate of a solid propellant is closely related to the pressure[37].

References

  1. Yilmaz, N.; Donaldson, B.; Gill, W.; Erikson, W. Solid propellant burning rate from strand burner pressure measurement. Propellants Explos. Pyrotech. 2008, 33, 109-117.
  2. Literature should include recent papers from Aerospace (2022, 9(6), 325; 2022, 9(11), 677; 2023, 10(1), 45)

Response: Thank you very much for your comment. References have been added to the introduction and text. The details are as following.

Nano-aluminum powder can reduce the ignition delay time and improve the energy release efficiency of the propellant[14,15].

References

  1. Liu, X.-L.; Hu, S.-Q.; Liu, L.-L.; Zhang, Y. Condensed combustion products characteristics of HTPB/AP/Al propellants under solid rocket motor conditions. Aerosp. 2022, 9, 677.
  2. Cha, J.; de Oliveira, É.J. Performance Comparison of Control Strategies for a Variable-Thrust Solid-Propellant Rocket Motor. Aerosp. 2022, 9, 325.
  3. Yilmaz, N.; Donaldson, B.; Gill, W. Aluminum Particle Ignition Studies with Focus on Effect of Oxide Barrier. Aerosp. 2023, 10, 45.
  4. Muravyev, N.; Frolov, Y.; Pivkina, A.; Monogarov, K.; Ordzhonikidze, O.; Bushmarinov, I.; Korlyukov, A. Influence of particle size and mixing technology on combustion of HMX/Al compositions. Propellants Explos. Pyrotech. 2010, 35, 226-232.
  5. There are several commercial codes which are mainly used for propellant combustion modeling. It is recommended that the authors elaborate on that.

Response: Thank you very much for your comment. I don't understand what the commercial code you proposed is. In this paper, there are no commercial codes for propellant combustion modeling.

  1. Uncertainty analysis should be performed with the inclusion of experimental data.

Response: Thank you very much for your comment. The uncertainties of the data are analyzed and the details are given below.

The burning rate increases from 5.391±0.021mm/s to 7.244±0.052mm/s at 7MPa of pressure; meanwhile, the pressure exponent of the burning rate law is changed from 0.326±0.047 to 0.483±0.045, and the d43 of the combustion residue is reduced from 165.31±36.18μm to 12.95±4.00μm.

  1. Graphs should have uncertainty bars.

Thank you very much for your comment. Figure 2 is redrawn, the details as detailed below.

Figure 2. Linear burning rate and power-law pressure dependence in 3MPa-9MPa for solid propellants.

  1. Conclusions somewhat repeat what was mentioned in the abstract. I recommend the authors include “future work” as well as the importance of this work in relation to practical applications.

Response: Thank you very much for your comment. The conclusion was amended. It is pointed out that aluminum-lithium alloys have potential applications in solid propellants.

Round 2

Reviewer 3 Report

The paper has been revised according to reviewers' comments. It is recommended for publication.